# Fibrosis Progression in Patients with Budd–Chiari Syndrome and Transjugular Intrahepatic Portosystemic Shunt (TIPS): A Long-Term Study Using Transient Elastography

**DOI:** 10.3390/diagnostics14030344

**Published:** 2024-02-05

**Authors:** Martin Rössle, Dominik Bettinger, Lukas Sturm, Marlene Reincke, Robert Thimme, Michael Schultheiss

**Affiliations:** 1Department of Medicine II, Medical Center University of Freiburg, Faculty of Medicine, University of Freiburg, Hugstetter Str. 55, 79106 Freiburg, Germany; dominik.bettinger@uniklinik-freiburg.de (D.B.); marlene.reincke@uniklinik-freiburg.de (M.R.); robert.thimme@uniklinik-freiburg.de (R.T.); michael.schultheiss@uniklinik-freiburg.de (M.S.); 2Berta-Ottenstein Programme, Faculty of Medicine, University of Freiburg, 79106 Freiburg, Germany

**Keywords:** Budd–Chiari syndrome, cirrhosis, fibrosis, TIPS, transient elastography, FibroScan

## Abstract

Hepatic vein outflow obstruction causes congestion of the liver, leading to necrosis, fibrosis, and portal hypertension (PH). A transjugular intrahepatic portosystemic shunt (TIPS) reduces congestion and PH by providing artificial outflow. The aim of the study was to investigate fibrosis progression in patients with Budd–Chiari syndrome (BCS) and TIPS using transient elastography (TE). From 2010 to 2022, 25 patients received 80 TEs using FibroScan^®^, Echosens, Paris, France (3.2 ± 2.1 per patient). TIPS function was assessed via Doppler ultrasound or radiological intervention. At the time of TE examination, 21 patients had patent shunts. Four patients had occluded shunts but normal pressure gradients during the intervention. The first TE measurement performed 9.8 ± 6.8 years after the BCS diagnosis showed stiffness values of 24.6 ± 11.5 kPa. A second or last measurement performed 7.0 ± 2.9 years after the first measurement showed similar stiffness values of 24.1 ± 15.7 kPa (*p* = 0.943). Except for three patients, the liver stiffness was always >12 kPa, indicating advanced fibrosis. Stiffness values obtained <5 years (*n* = 8, 23.8 ± 9.2 kPa) or >5 years after the BCS diagnosis (24.9 ± 12.7 kPa) did not differ (*p* = 0.907). In addition, stiffness was not related to the interval between BCS and TIPS implantation (*p* = 0.999). One patient received liver transplantation, and two patients died from non-hepatic causes. Most patients developed mild to moderate cirrhosis, possibly during the early phase of the disease. Timing of TIPS did not influence fibrosis progression. This and the release of portal hypertension may argue in favor of a generous TIPS implantation practice in patients with BCS.

## 1. Introduction

Primary Budd–Chiari syndrome (BCS) is characterized by thrombotic outflow obstruction, which involves the hepatic veins with or without extension into the inferior caval vein. The clinical presentation of the disease may vary from severe acute liver failure to asymptomatic chronic liver disease, depending on the extent and velocity of thrombosis development and collateral formation. Medical treatments such as anticoagulation and cytoreductive therapy are essential to prevent thrombus extension [1]. However, they are often ineffective with respect to recanalization of the hepatic veins and 80–90% of patients do not show a clear and durable clinical response [2]. In contrast, the congestive state may be abbreviated and the damage reduced by interventional treatment such as angioplasty in case of a short segment (web-like) BCS or portosystemic side-to-side shunts, such as the transjugular intrahepatic portosystemic shunt (TIPS), which allows transition of the portal vein into an outflow, thereby improving hepatic perfusion [3,4,5].

Transient elastography (TE) evaluates fibrosis by determining the liver stiffness. It is a recommended alternative to liver biopsy in patients with chronic viral hepatitis, steatotic liver disease, and alcoholic hepatitis [6,7,8]. It has also been applied to patients with BCS to evaluate short-term changes after hepatic vein angioplasty [9,10,11,12].

Patients with BCS may also develop cirrhosis. However, little is known about the progression of fibrosis and effects of TIPS treatment. Our study provides data on the long-term development of liver fibrosis/cirrhosis using repeated TE measurements in patients with long-segment BCS who underwent TIPS implantation.

## 2. Methods

### 2.1. Patient Selection and Study Design

From 1992 to 2022, 109 patients with symptomatic BCS were treated at the University Medical Center, Freiburg. The study was approved by the local ethics committee (23-1209-S1-retro) and is in accordance with the current version of the Declaration of Helsinki. This observational study was part of the Freiburg TIPS Registry (NCT05782556). Owing to the retrospective nature of the study, the need for informed patient consent was waived.

All patients were screened for pre-existing liver diseases, such as hepatitis B and hepatitis C virus infection, alcoholic liver disease, and metabolic or hereditary liver diseases. Anticoagulation therapy with heparin or low molecular weight heparin was initiated immediately after a diagnosis of BCS was established. TIPS implantation together with continuation of anticoagulation therapy was performed in 95 patients (87%) who failed medical treatment, which was defined as persistent tense ascites and progressive renal and/or liver failure [13]. Since the acquisition of FibroScan^®^ (Echosense, Paris, France) in 2010, 25 consecutive and unselected TIPS outpatients received TE measurements and were included in this observational study. The intervals between TE measurements were not predetermined because the visit intervals differed according to individual needs.

### 2.2. TIPS Implantation

The detailed technique of TIPS implantation has been previously described [4,5,14]. The right transjugular approach was used in all patients, and a puncture needle was advanced into the inferior caval vein. Puncture of the portal vein via the hepatic vein or by direct transcaval puncture was always performed using ultrasound guidance. After portography and measurement of the portal and right atrial pressures, the parenchymal tract was dilated and a stent graft was placed. The final portography and pressure measurements were performed. Seven patients underwent the implantation before covered stents were available. They received covered stents (Viatorr^®^ or ViatorrCX^®^, W.L. Gore & Associates, Putzbrunn, Germany) only during the revision. Except for one patient who received a Viatorr stent with a fixed diameter of only 8 mm, all patients received wider stents with diameters of 10–12 mm.

### 2.3. Colour-Coded Duplex Sonography (CDUS) and TE Measurement

During follow-up, patients underwent CDUS and, since 2010, TE measurements. All examinations were performed by experienced physicians (MR, MS). CDUS was performed using a Logic E9 sonographic unit (GE Healthcare, Solingen, Germany). Impaired shunt function was suspected if the portal vein flow velocity was <25 cm/s and stent flow velocity was <40 cm/s or >180 cm/s [14]. These patients, as well as those showing clinical signs of portal hypertension (redevelopment of ascites and/or varices > grade I), were allocated to shunt revision. At the time of TE measurement, duplex sonography was performed to exclude shunt failure and all patients were free of ascites or of any other clinical sign of significant portal hypertension. TE measurements were performed under a 4 h fasting condition during breath-holding. The right arm was placed in abduction and the appropriate intercostal space was located sonographically. At least 10 examinations at breath-hold were performed and expressed as mean ± SD. TE measurements were interpreted according to the European Association for the Study of the Liver (EASL) clinical practice guidelines [15]: advanced fibrosis (>stage 3–4), defined by the Metavir histology scoring system [16], is very probable with TE values above 10–15 kPa, and clinically significant portal hypertension due to cirrhosis is very likely (about 90%) with values exceeding 20 kPa.

### 2.4. Statistical Analyses

Continuous variables are reported as means with standard deviations and medians with minimum and maximum values. Categorical variables are reported as frequencies with percentages. Group differences were determined using the chi-squared test for categorical variables. As there was no Gaussian distribution of the data, Mann–Whitney U tests or Wilcoxon rank-sum tests were used as appropriate. Multivariate regression analysis (ANOVA) was used to assess predictors of liver stiffness. Therefore, the mean value of all measurements was used as the dependent variable. The predictive value of baseline data before TIPS implantation was assessed. A *p*-value of <0.050 was considered statistically significant. Statistical analyses were performed using SPSS (version 27.0, IBM, New York, NY, USA), GraphPad Prism (version 10, GraphPad Software, San Diego, CA, USA), and STATA (version 17.0, Stata Corp Lp., College Station, TX, USA).

## 3. Results

### 3.1. Patients’ Characteristics

The clinical characteristics of the 25 patients are summarized in Table 1. All patients had acute or symptomatic disease at the time of BCS diagnosis. Female sex (76%) and myeloproliferative disease (76%) were predominant. Preexisting liver diseases were excluded at the time of BCS diagnosis. All patients received anticoagulation and/or cytoreduction therapy after BCS diagnosis. All patients had stable liver function throughout the follow-up observation period, and no clinically overt hepatic encephalopathy was detected. Patient #24 (Table 2 and Table 3) developed slowly progressive liver failure and underwent liver transplantation 2.8 years after the diagnosis of BCS and 2.7 years after TIPS implantation. Patient #2 and patient #7 (69 and 79 years old) died 23.5 years and 22.2 years after onset of the BCS from cardiac failure after valve replacement operation and from osteomyelofibrosis, respectively.

### 3.2. Timing of TIPS Treatment

Table 2 summarizes the timing of TIPS treatment. The interval from the diagnosis of BCS to TIPS implantation was 430 ± 680 days (median 104 days; range 3 days–6.2 years). Ten patients (40%) underwent TIPS implantation within one month of the acute phase of the disease. The other patients had a subacute or chronic disease at the time of TIPS implantation. The mean follow-up was 15.2 ± 7.6 years (17 years; 1.6–27.5 years) and 14.1 ± 7.2 years (15.9 years; 1.6–24.1 years) after the diagnosis of BCS and after the TIPS implantation, respectively.

**Table 2 diagnostics-14-00344-t002:** Date of diagnosis of Budd–Chiari syndrome, time from diagnosis to TIPS treatment (days), and follow-up time from diagnosis to TIPS implantation (years).

Patient Number	Date of BCS Diagnosis	Time from Diagnosis to TIPS (Days)	Follow-Up after Diagnosis (Years)	Follow-Up after TIPS (Years)
1	1995	1699	27.5	22.8
2	1995	109	23.5	23.2
3	1997	2161	24.6	18.7
4	1998	14	24.1	24.1
5	1999	443	20.6	19.4
6	1999	578	23.9	22.3
7	2000	11	22.2	22.2
8	2000	14	22.5	22.4
9	2001	334	18.2	17.3
10	2004	10	17.3	17.3
11	2004	639	17.7	15.9
12	2005	18	13.4	13.3
13	2005	24	17.8	17.7
14	2006	104	16.9	16.7
15	2007	3	13.4	13.4
16	2008	2265	14	7.8
17	2008	72	14	13.8
18	2010	1271	11.8	8.3
19	2012	516	7.9	6.5
20	2013	212	9.7	9.1
21	2015	34	3.2	3.1
22	2016	168	6.8	6.3
23	2017	10	5.8	5.8
24	2019	28	2.7	2.6
25	2022	7	1.6	1.6
mean ± SD (median; range)		430 ± 680	15.2 ± 7.6	14.1 ± 7.2
		(104; 3–2265)	(17.0; 1.6–27.5)	(15.9; 1.6–24.1)

### 3.3. Shunt Patency

Shunt patency was controlled using CDUS immediately before TE examinations were performed. Primary shunt patency was observed in four patients. Twenty-one patients required a total of 49 revisions, with 2.0 ± 2.0 revisions per patient (1; 1–8). Patency was restored in 17 patients by stent-in-stent implantation or the placement of a parallel stent (*n* = 2). In four patients, radiological revision revealed pressure gradients below 12 mmHg, the reason why stent patency was not restored. Two characteristic examples are shown in Figure 1. In addition, sonographic examinations excluded ascites, hepatocellular carcinoma, and portal vein thrombosis.

### 3.4. TE Measurements

As shown in Table 3, the interval from diagnosis to first TE measurement was 9.8 ± 6.8 years (median 8.8 years) ranging from 0.59 (214 days) to 20 years. The second/last TE measurement, which was available in only 17 patients, was performed 7.0 ± 2.9 years after the first measurement (7.0 years; 1–12.4 years). First and last measurements were 24.1 ± 11.5 kPa (21.3 kPa; 6.2–49.6 kPa) and 24.1 ± 15.7 kPa (20.5 kPa; 7.5–66.4 kPa), respectively, without showing a difference (*p* = 0.962). The mean of all longitudinal measurements was 26.0 ± 13.0 kPa (22.9 kPa; 7.2–50.7 kPa) without showing a significant difference from the first measurement (*p* = 0.124). Except for three patients (patients #3, #10, and #17), the liver stiffness was always >12 kPa, indicating advanced fibrosis. In two patients (patients #24 and #25), early measurements performed 0.59 years (214 and 217 days) after the diagnosis of BCS already showed stage IV fibrosis, suggesting that fibrosis may develop during the acute or subacute phase. Accordingly, measurements performed <5 years after the diagnosis (*n* = 8, 23.8 ± 9.2 kPa; 25.5 kPa; 12.8–40 kPa) did not differ from those performed >5 years after the diagnosis (24.9 ± 12.7 kPa; 21.3 kPa; 6.2–49.6 kPa, *p* = 0.907).

**Table 3 diagnostics-14-00344-t003:** Time from diagnosis to first TE measurement, first and last FibroScan measurements, interval in years between the first and last measurements, as well as individual means of longitudinal measurements and number of measurements performed during follow-up.

Patient Number	Time from Diagnosis of BCS to First FibroScan (Years)	First FibroScan, Mean of 10 Measurements, (kPa)	Last FibroScan, Mean of 10 Measurements (kPa)	Interval between First and Last FibroScan (Years)	Mean of All Longitudinal FibroScan Measurements (kPa)	Number of Measurements (*n*)
1	18	21.3	20.5	9.5	22.9	3
2	18.6	27	37.4	4.9	32.2	2
3	12.7	9.5	9.7	12	10.4	6
4	15.9	35.3	20	7.7	39.4	4
5	14.7	27	14.5	4.5	17.7	5
6	20	44.9			44.9	1
7	15.2	14.5	21.1	6	20.2	5
8	18.9	49.6	38	4.2	49.6	1
9	18.3	42			42	1
10	7.3	6.2	7.5	9.1	7.2	6
11	9.8	16.5			16.5	1
12	8.2	12.6	13.5	5.2	12.2	5
13	8.8	36.3	49.6	9	45.2	3
14	16.7	18.7			18.7	1
15	12.9	18.5			18.5	1
16	7	22.1	21.3	7	29	4
17	1.6	14.1	12	12.4	10.1	7
18	3.5	28	66.4	8.3	50.7	7
19	7.9	21.3			21.3	1
20	2.4	40	21.6	7.3	29.5	4
21	1.2	23.4	28.4	2	26.5	3
22	3.5	12.8	12.1	1	12.5	2
23	1.2	15.9	15.7	4.7	15.2	5
24	0.6	27.6			27.6	1
25	0.6	28.8			28.8	1
Mean ± SD	9.8 ± 6.8	24.1 ± 11.5	24.1 ± 15.7	7.0 ± 2.9	26.0 ± 13.0	3.2 ± 2.1
(median, range)	(8.8; 0.6–20)	(21.3; 6.2–49.6)	(20.5; 7.5–66.4)	(7.0; 1.0–12.4)	22.9; 7.2–50.7	(3; 1–7)

The individual values of the first and last measurements are shown in Figure 2. With a few exceptions, the variations between the two measurements were moderate. The results of the individual longitudinal measurements performed over time are shown in Figure 3. Except for one patient (patient #18), longitudinal measurements did not show relevant deviations from the first measurement or transition from cirrhosis to fibrosis stages I–III or vice versa.

The effect of TIPS implantation on fibrosis progression was assessed by comparing the liver stiffness values of patients receiving TIPS in the early phase (within one month from the beginning of symptomatic disease: *n* = 10) with those of patients receiving delayed TIPS implantation (after one month, *n* = 15). The mean of all longitudinal measurements after early and delayed TIPS treatment was 28.1 ± 15.5 kPa (23.8 kPa; 7.2–49.6 kPa) and 25.7 ± 12.5 kPa (22.9 kPa; 10.1–50.7 kPa), respectively, without showing a significant difference (*p* = 0.999). It is important to emphasize that disease severity differed significantly between the groups, disfavoring the group receiving early treatment (Table 4). Aspartate aminotransferase (AST), international normalized ratio (INR), white blood cells, and prognostic Clichy and Rotterdam scores [17,18], but not Child-Pugh and MELD scores, were significantly different between the groups. The elevated INR in the group receiving early TIPS is caused by the acute liver disease and not related to anticoagulation.

As demonstrated in the multivariate regression analysis (Table 5), stiffness did not depend on age, pre-TIPS baseline biochemical variables, or prognostic scores, as depicted in Table 4.

## 4. Discussion

Hepatic outflow obstruction causes congestion of the liver and may result in parenchymal extinction, nodular hyperplasia, and eventually cirrhosis [19,20,21]. In addition, portal hypertension may occur early or late owing to congestion or cirrhosis, respectively. Treatment with TIPS improves and prevents early or late portal hypertension. It releases congestion by providing an alternative route backward through the hepatic portal branches and shunt [4]. This may be the reason for the rapid improvement in liver enzymes in patients with acute or fulminant BCS receiving TIPS [5]. A positive long-term effect of surgical shunts on fibrosis progression has been previously suggested [21,22], but a meaningful study is missing.

Measurement of liver stiffness by transient elastography (TE) using FibroScan or shear wave elastography (SWE) is a non-invasive method used to evaluate the fibrosis stage in patients with various liver diseases [6,7,8]. It has recently been applied to investigate the effect of endovascular intervention on liver stiffness and congestion in patients with short-length BCS [9,10,11,12]. Our study investigated the long-term development of liver stiffness in patients with complete BCS who received TIPS.

Most of our patients (92%) had liver stiffness values that indicated cirrhosis (>12 kPa). Stiffness was stable over intervals between measurements of 1.0 to 12.4 years, with similar values obtained at the beginning or end of the study. Patients receiving measurements earlier (<5 years from diagnosis) or later (>5 years from diagnosis) showed comparable stiffness values, assuming that stiffness was stable over time and developed at the beginning of the disease. This view is supported by the finding of stage 4 fibrosis in two patients who underwent stiffness measurement 214 and 217 days after the onset of the disease. The time between diagnosis of BCS and TIPS implantation was not related to stiffness. Patients receiving TIPS treatment early (within 30 days of disease onset) or delayed showed similar stiffness values. This is in agreement with a previous study [17], but does not confirm the surgical studies pointing towards a positive effect of surgical shunting on fibrosis development [21,22]. However, patients with early TIPS have more serious disease, potentially accelerating fibrosis development. Thus, the positive effects of early TIPS treatment in patients with acute and severe diseases cannot be excluded. However, with the exception of one patient requiring liver transplantation, TIPS did not aggravate liver disease or worsen hepatic function, and no patient died because of a liver-related reason. In addition, no patients developed clinically overt hepatic encephalopathy or refractory complications of portal hypertension.

Stiffness did not depend on age, pre-TIPS baseline biochemical variables, or prognostic scores. This suggests that, with respect to fibrosis progression, prognostic scores are not useful in BCS patients receiving or having received TIPS implantation.

The main factors contributing to liver stiffness in patients with BCS are liver congestion and fibrosis. Three studies, including 89 patients with short-segment BCS, have measured liver stiffness before and after decongestion by endovascular intervention [9,10,11]. Except for four patients who received TIPS [11], all patients underwent angioplasty with or without stent implantation in the hepatic or inferior caval veins. Liver stiffness was determined using shear wave elastography (SWE) in two studies [9,11] and TE (FibroScan^®^) in one study [10]. The remaining study included 13 patients and used magnetic resonance imaging to determine liver stiffness in various segments of the liver [12]. In all studies, the intervention resulted in a marked immediate decrease in liver stiffness by 32–58%. This was uniformly ascribed to decongestion of the liver by angioplasty. Accordingly, two of the three studies measuring pressure found a correlation between the decrease in the pressure gradient across the stenotic hepatic vein and the decrease in liver stiffness [9,12].

Liver biopsies were performed during the intervention. Surprisingly, they did not find a correlation between the Metavir fibrosis score and liver stiffness before or after the intervention [9,10,11]. In addition, the fibrosis scores obtained by TE were much higher than the corresponding Metavir fibrosis scores [10,11]. For instance, patients with Metavir fibrosis stage ≤ 2 showed stiffness values of 31.16 kPa indicating fibrosis stage 4, whereas patients with Metavir fibrosis stage > 2 showed lower TE stiffness values of 25.36 kPa [10]. It can be assumed that many patients had additional small hepatic vein obstruction, which affected stiffness, although the large hepatic veins were patent after angioplasty. As also demonstrated, mean stiffness after angioplasty was still increased and amounted to between 20 and 30 kPa, independently [9,10,11]. Presumably, patients with a lower fibrosis stage had higher residual congestion, and those with a greater stage of fibrosis had lower or no congestion, with little or no contribution to stiffness. This interpretation may be substantiated by the assumption that a lower fibrosis stage may be related to shorter follow-up, insufficient collateralization, and greater congestion, and vice versa. Unfortunately, detailed information on the interval between the diagnosis of BCS and angioplasty/biopsy has not been presented [9,10,11]. Measurement of the wedge hepatic vein or portal pressures that were not performed would not answer this question, since these measurements cannot distinguish between post-sinusoidal congestion influencing stiffness and portal hypertension due to fibrosis not influencing stiffness. The different effects on the stiffness may be explained by the locations of the resistances. Congestion is caused by a post-sinusoidal block leading to parenchymal/sinusoidal swelling, while cirrhotic portal hypertension may be the result of a more peripheral (portal) location of the resistance that does not cause swelling and only slightly influences stiffness [23,24,25].

Based on these considerations, stiffness measurement is not a reliable tool for determining fibrosis stage in unselected patients with BCS.

In contrast to the studies mentioned above, our patients had chronic disease, and congestion influencing the stiffness measurement was excluded. All patients had functioning TIPS or large collaterals, replacing the occluded TIPS. In these latter patients, congestion was excluded due to a lack of signs of portal hypertension and radiological verification of a portosystemic pressure gradient of <12 mmHg. Thus, in contrast to the aforementioned studies [9,10,11], our stiffness measurements may reliably reflect the fibrosis stages. Certainly, validating TE in patients with BCS may be desired. However, the rarity of the disease and ethical reasons raise doubts about conducting such a study in patients with chronic BCS and TIPS.

Our results may have been limited by the uneven distribution of fibrosis in the liver. However, this is unlikely because all our patients had complete BCS involving all three hepatic veins. This was clearly verified by imaging and portography during TIPS implantation, showing a lack of perfusion of all hepatic segments or lobes. Other factors influencing stiffness [26,27], such as severe inflammation of the liver or right-sided heart failure, can also be excluded. Our study lacks stiffness measurements before TIPS implantation, which may indicate that congestion was possibly resolved by TIPS. A recent study including seven patients performed liver stiffness measurements before and after TIPS in two patients with a patent shunt [28]. Both showed a reduction in stiffness from 75 kPa before TIPS to 8 kPa and 7.3 kPa after TIPS, respectively, confirming the complete release of congestion by TIPS. The limited sample size of our study is due to the long observation time, which caused some fluctuation in patients coming from all over Germany and abroad. The patients were not selected, and duplex and TE measurements were performed by experienced investigators.

In conclusion, TE follow-up of patients with BCS treated with TIPS showed stable long-term disease and excellent liver-related prognosis, although most patients developed mild cirrhosis during the early phase of the disease. TIPS did not deteriorate or improve the development of fibrosis. In accordance with previous suggestions [4,29], the measurements presented and the safe prevention of clinical complications of portal hypertension may question the current treatment recommendations and favor a generous TIPS implantation practice in patients with BCS. A multicenter study is recommended including stiffness measurements before and after TIPS, TE and wedge pressure measurements at standardized timings, preferably together with transjugular liver biopsy, and a control group without TIPS.

## Figures and Tables

**Figure 1 diagnostics-14-00344-f001:**
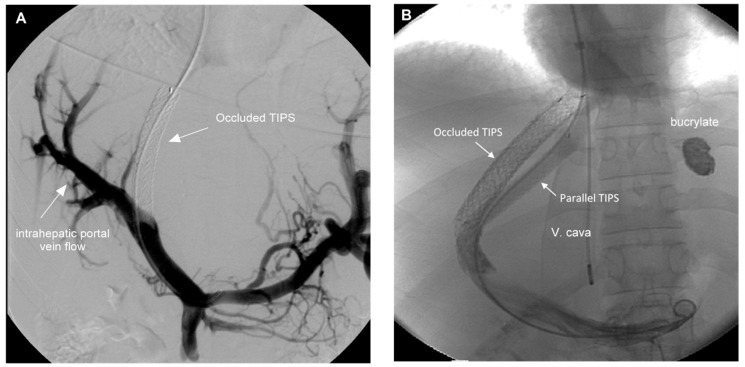
(**A**). Portography of patient #17 five months after TIPS implantation. CDUS showed occluded TIPS in the absence of clinical signs of portal hypertension. The angiogram demonstrates good intrahepatic portal perfusion and the pressure gradient was 10 mmHg. The shunt was not restored and the catheter removed. (**B**). Portography of a 16-year-old female patient (#4) with fulminant BCS in 1998 requiring early TIPS implantation. After seven revisions between 1998 and 2013 a serious variceal bleeding from gastric varices occurred. The catheterization of the occluded stent-shunt was not possible and a transcaval puncture was performed to implant a parallel stent. The shunt has been fully patent since 2013 and the patient gave birth to a healthy child in 2019.

**Figure 2 diagnostics-14-00344-f002:**
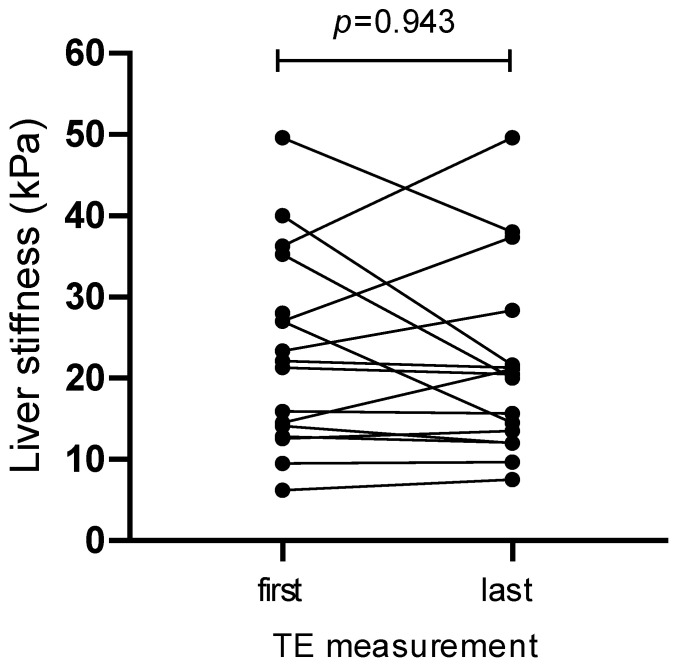
Individual first (*n* = 25) and last (*n* = 17) liver stiffness measurements (kPa: kilo-Pasqual).

**Figure 3 diagnostics-14-00344-f003:**
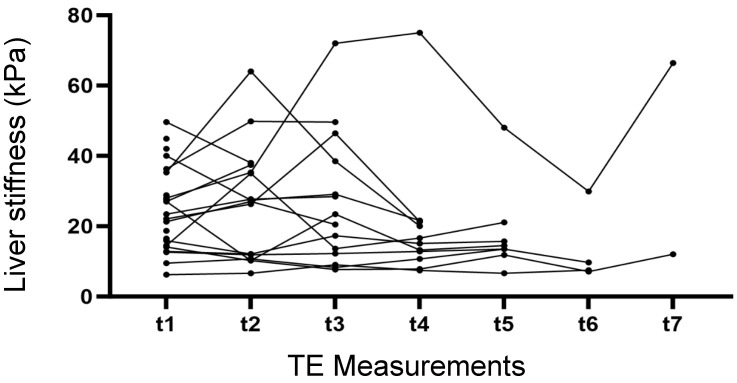
Longitudinal liver stiffness measurements of individual patients over time.

**Table 1 diagnostics-14-00344-t001:** Clinical characteristics of the 25 patients at baseline (before TIPS) and at follow-up.

Age at Diagnosis of BCS, Mean ± SD (Median; Range)	39.2 ± 12.7 (39.0; 16–63)
Female gender, *n* (%)	19 (76)
Etiology, *n*	
-Myeloproliferative syndrome	19
-Paroxysmal nocturnal hemoglobinemia	1
-Antiphospholipid syndrome	1
-Eosinophilic syndrome	1
-Unknown	3
Prognostic variables before TIPS, mean ± SD (median; range)	
-Child-Pugh score	8.6 ± 2.4 (8; 6–15)
-MELD score	14.3 ± 7.5 (12.5; 7–38)
-Clichy score	5.8 ± 1.1 (5.8; 3.5–8.8)
-Rotterdam score	2.4 ± 0.76 (2.0; 1.7–3.9)
Specific medication during follow-up, *n*	
-Phenprocoumon	14
-DOAC	10
-Acetylicsalicilic acid	5
-Interferon	2
-Ruxolitinib	3
-Hydroxyurea	1
Biochemical tests at end of follow-up, mean ± SD (median; range)	
-Bilirubin, mg/dL	1.2 ± 0.5 (1.0; 0.5–2.4)
-Aspartate-Aminotransferase (AST), U/L	32.5 ± 9.0 (34.0; 17–47)
-Alanine-Aminotransferase (ALT), U/L	32.0 ± 23.4 (27; 9–101)
-y-Glutamyltransferase (yGT), U/L	108.8 ± 121.5 (73.5; 25–645)
-Alkaline Phosphatase, U/L	130.0 ± 72.0 (106; 53–367)
-Hemoglobin, g/dL	12.7 ± 2.2 (13.0; 7.2–16.4)
-Platelet count, ×10^3^	298 ± 180 (290; 73–778)
Clinical signs of portal hypertension at stiffness measurements	
-No Ascites	25
-Varices at last follow-up	
-no	19
-grade 1	6
Liver transplantation	1
Death	2

**Table 4 diagnostics-14-00344-t004:** Age and baseline (before TIPS implantation) biochemical tests and prognostic scores of patients receiving early (within 30 days) or delayed TIPS treatment after the clinical onset of Budd–Chiari syndrome. Values are the mean ± SD (median, range).

Variable	Early TIPS *n* = 10	Delayed TIPS *n* = 15	Significance (*p*)
Age (years)	39.7 ± 13.5 (40.8; 16.1–58.3)	38.9 ± 12.6 (39.0; 17.3–63.3)	0.868
Bilirubin (mg/dL)	3.1 ± 3.6 (1.6; 0.6–11.6)	1.6 ± 0.7 (1.5; 0.5–3.1)	0.663
Aspartate-Aminotransferase (AST, U/L)	551 ± 635 (370; 29–2110)	35 ± 27 (29; 14–101)	0.001
Lactate-Dehydrogenase (LDH, U/L)	615 ± 723 (370; 120–2476)	285 ± 165 (228; 164–663)	0.074
International normalized ratio (INR)	2.0 ± 0.9 (1.6;1.2–4)	1.4 ± 0.4 (1.3; 1–2.7)	0.020
White blood cells (10^9^/L)	19.1 ± 7.9 (21; 8.4–31)	8.8 ± 3.3 (8.4; 5.5–18.7)	0.001
Platelet count ×10^3^	233 ± 140 (201;73–520)	343 ± 196 325;112–778)	0.219
Creatinine (mg/dL)	1.2 ± 0.8 (0.9; 0.6–3.3)	1.0 ± 0.4 (0.9; 0.5–2.1)	0.074
Child-Pugh score	9.3 ± 2.3 (10; 6–12)	8.1 ± 2.3 (7.5; 6–11)	0.221
MELD score	17.4 ± 10.1 (14; 8–38)	12.2 ± 4.3 (11; 7–22)	0.190
Clichy score	6.5 ± 1.0 (6.3; 5.2–8.8)	5.4 ± 1.0 (5.2; 3.5–7.1)	0.032
Rotterdam score	2.8 ± 0.9 (2.2; 1.8–3.9)	2.1 ± 0.5 (1.9; 1.7–3.3)	0.030

**Table 5 diagnostics-14-00344-t005:** Multivariate regression analysis of baseline variables obtained at diagnosis of BCS and before the TIPS intervention with respect to liver stiffness.

Variable	Standardized Coefficient β	95% CI	*p* Value
Bilirubin	0.12	−3.74–5.03	0.756
Creatinine	−0.16	−18.24–11.14	0.612
INR	−0.36	−23.10–10.43	0.431
Clichy	0.46	−7.82–18.24	0.405
Rotterdam	−0.23	−21.5–13.71	0.642
White cell count	−0.29	−1.65–0.65	0.370
Age	−0.04	−0.95–0.87	0.930

## Data Availability

The data presented in this study are available on reasonable request from the corresponding author.

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
