# Peer review of "Fibrosis Progression in Patients with Budd–Chiari Syndrome and Transjugular Intrahepatic Portosystemic Shunt (TIPS): A Long-Term Study Using Transient Elastography"

_diagnostics, 2024, doi:10.3390/diagnostics14030344_

Round 1
Reviewer 1 Report
Comments and Suggestions for Authors
In general, the manuscript is well written and of clinical interest. However, authors need to address some point:
- Add in the methods the fibrosis degree in relation to kPa values.
- Figure legend should be mentioned under the figure.
- Correct the spelling of alkaline phosphatase in table 1
- In table 4: clarify if the patients with early TIPS were on anti-coagulation or not (as mean INR was 2). In the same table add to the comparison platelets count as a marker of portal hypertension. In addition, please add to this table comparison between results of last Fibroscan in both groups.
- In the discussion section you mentioned that:''Liver biopsies were performed during the intervention. Surprisingly, they did not find a correlation between the metavir fibrosis score and liver stiffness before or after the intervention''. Clarify if the fibrosis detected by liver biopsy was better or worse than that detected by Fibroscan.
- Did the presence of ascites obscured the field for proper measurement of stiffness by fibroscan?
- The manuscripts needs grammatical and editing revision.
Comments on the Quality of English LanguageNeeds some grammar editing
Author Response
Thank you very much for your valuable comments. All of them have been addressed and highlighted in the manuscript in yellow.
- Fibrosis degree in relation to kPa values is now incorporated on page 3, L 93: TE measurements were interpreted according to the European Association for the Study of the Liver (EASL) clinical practice guidelines (15): Advanced fibrosis (> stage 3-4), defined by the Metavir histology scoring system (16), is very probable with TE values above 10-15 kPa, and clinically significant portal hypertension due to cirrhosis is very likely (about 90%) with values exceeding 20 kPa. The respective references (15, 16) are also added.
- Text, tables and figures are rearranged for better readability. Legends are added below figures.
- Spelling of alkaline phosphatase is changed as suggested.
- The value of the INR in table 4 is now commented in the text (page 9, L 226): The elevated INR in the group receiving early TIPS is caused by the acute liver disease and not related to anticoagulation. The fibroscan results were not included because the table represents only baseline values. The results of the fibroscans are given in the text (page 9, L 219).
- The relation of stiffness measured by TE and the Metavir histological score is now presented on page 11, L 292: In addition, the fibrosis scores obtained by TE before or early after TIPS were much higher than the corresponding Metavir fibrosis scores (10, 11). For instance, patients with Metavir fibrosis stage ≤2 showed stiffness values after TIPS of 31.16 kPa indicating fibrosis stage 4, whereas patients with Metavir fibrosis stage >2 showed lower TE stiffness values of 25.36 kPa (10).
- Patients with ascites were excluded. They received TIPS revision and TE measurement was performed after reabsorption of the ascites (see page 3, L 86): These patients, as well as those showing clinical signs of portal hypertension (redevelopment of ascites and/or varices > grade I), were allocated to shunt revision. At the time of TE measurement, duplex sonography was performed to exclude shunt failure and all patients were free of ascites or of any other clinical sign of significant portal hypertension.
- Language edition has been performed previously.
Reviewer 2 Report
Comments and Suggestions for Authors
The aim of the study was to investigate fibrosis progression in patients with Budd-Chiari syndrome and TIPS using transient elastography. This study used transient elastography to measure liver stiffness over time in 25 patients with Budd-Chiari syndrome who were treated with TIPS. The results showed that most patients developed mild to moderate cirrhosis early in the disease course, and liver stiffness remained stable over long-term follow-up without worsening fibrosis. Timing of TIPS placement did not affect fibrosis progression. The findings suggest that TIPS releases congestion and prevents complications of portal hypertension without aggravating liver fibrosis in Budd-Chiari syndrome. The authors conclude that these results support a more generous practice of TIPS implantation for patients with this condition. In summary, the authors did tackle some of the limitations but a multi-center, controlled study with pre-TIPS elastography remains ideal to address the cons more fully.
The Discussion shows insight into the potential weaknesses. The authors acknowledge certain limitations related to sample size, uneven measurements, lack of baseline, pre-TIPS data, proving temporal causation, accounting for confounders for liver stiffness measurements, generalizability, and lack of comparators comparing treatments. Specifically, they attribute the small sample size to long observation time and fluctuating patients. They recognize the uneven distribution of measurements. They note the lack of elastography prior to TIPS to assess decongestion. Without definitive claims about causation, they are careful in relating TIPS to fibrosis outcomes. They discuss and exclude some confounding factors potentially influencing stiffness. While not addressed directly, the single-center design limits generalizability. Lastly, no comparison of treatments was made since this was an observational study. However, the authors show insight into these potential weaknesses in the Discussion. Ultimately, larger multi-center controlled trials with pre-TIPS elastography are needed to further confirm and expand on these initial findings.
Although sample size is limited, here are some suggestions on areas for efforts if feasible:
Study Design:
- Include a control group without TIPS for comparison of fibrosis progression.
- Conduct study across multiple centers to improve sample size and generalizability.
- Standardize the timing and number of elastography measurements.
- Obtain elastography data before and after TIPS to assess effects on congestion.
- Expand inclusion criteria beyond just TIPS patients to increase applicability.
- Increase follow-up time and long-term outcomes data.
Methodology:
- Provide more details on patient selection criteria and demographics.
- Use liver biopsy in a subset of patients to validate elastography findings.
- Account for potential confounding factors affecting liver stiffness.
- Provide hemodynamic data such as portal pressures pre- and post-TIPS.
Analysis:
- Use multivariate analyses to control for confounders.
- Correlate elastography with lab values, symptoms, and clinical outcomes.
- Compare stiffness between TIPS placement time groups.
Comments on the Quality of English Language
none.
Author Response
Thank you very much for your valuable comments. Unfortunately, our data do not allow to address all of your suggestions.
- Study design
We accept that a control group as well as the inclusion of multiple centers would improve the manuscript considerably. However, as to our knowledge, such data are not available. Standardization of the timing of measurements and measurements before and after TIPS should be addressed in future multicenter trials. A respective statement has been introduced at the end of the discussion and highlighted in blue (page 12, L 344): A multicenter study is recommended including stiffness measurements before and after TIPS, TE and wedge pressure measurements at standardized timings preferably together with transjugular liver biopsy, and a control group without TIPS.
- Methodology
- We studied consecutive outpatients. A selection was not performed. Demographics are given in table 1.
- In this retrospective study, liver biopsies have not been performed. We agree that biopsies were very informative and worth to be taken. However, transjugular biopsies in patients requiring a TIPS revision may be technically challenging because appropriate liver veins are lacking. The need for a transcaval puncture increases the risk of the biopsy particularly in view of ongoing anticoagulation.
- Confounding factors affecting stiffness are excluded by the patients´ history, clinical and biochemical tests and discussed (page 11, L 327).
- We did not provide data on pressures pre- and post-TIPS because these data may not be of relevance during long-term follow-up.
- Analysis
- A regression analysis of baseline values (before TIPS) and stiffness (kPa) has been performed (see table 5). Biochemical variables and specific BCS prognostic scores (Vichy, Rotterdam, Child, MELD) are included in the analysis. No correlation was seen between variables and stiffness.
- Comparison of stiffness values in different TIPS placement groups is presented on page 9 and table 4. Liver stiffness in patients receiving early TIPS (< 30 days after onset of the BCS) was very similar to those receiving delayed TIPS (>30 days). See page 9, L 219): “The mean of all longitudinal measurements after early and delayed TIPS treatment was 28.1±15.5 kPa (23.8 kPa; 7.2-49.6 kPa) and 25.7±12.5 kPa (22.9 kPa; 10.1-50.7 kPa), respectively, without showing a significant difference (p=0.999).and
page 6, L 174: “measurements performed <5 years after the diagnosis (n=8, 23,8±9,2 kPa; 25.5 kPa; 12.8-40 kPa) did not differ from those performed >5 years after the diagnosis (24,9±12,7 kPa; 21.3 kPa; 6,2-49.6 kPa, p=0.907)”.
Round 2
Reviewer 1 Report
Comments and Suggestions for Authors
All comments have been addressed. The manuscript is now proper for publication.
Reviewer 2 Report
Comments and Suggestions for Authors
The major recurring themes are the lack of a control group, need for biopsy-correlation, accounting for confounders, and providing more analytical details. Addressing these with the addition of pre-TIPS data and optimized study design would substantially improve future study's impact and contribution.
Comments on the Quality of English Languagenone.